# Subcortical evidence for a contribution of arousal to fMRI studies of brain activity

Xiao Liu[1,2,3], Jacco A. de Zwart[1], Marieke L. Schölvinck[4], Catie Chang[1], Frank Q. Ye[5], David A. Leopold[5,6] & Jeff H. Duyn[1]

Cortical activity during periods of rest is punctuated by widespread, synchronous events in both electrophysiological and hemodynamic signals, but their behavioral relevance remains unclear. Here we report that these events correspond to momentary drops in cortical arousal and are associated with activity changes in the basal forebrain and thalamus. Combining fMRI and electrophysiology in macaques, we first establish that fMRI transients co-occur with spectral shifts in local field potentials (LFPs) toward low frequencies. Applying this knowledge to fMRI data from the human connectome project, we find that the fMRI transients are strongest in sensory cortices. Surprisingly, the positive cortical transients occur together with negative transients in focal subcortical areas known to be involved with arousal regulation, most notably the basal forebrain. This subcortical involvement, combined with the proto-typical pattern of LFP spectral shifts, suggests that commonly observed widespread variations in fMRI cortical activity are associated with momentary drops in arousal.

[1] Advanced MRI Section, Laboratory of Functional and Molecular Imaging, National Institute of Neurological Disorders and Stroke, National Institutes of Health, Bethesda, MD 20892, USA. [2] Multimodal and Computational Neuroimaging Laboratory, Department of Biomedical Engineering, The Pennsylvania State University, Pennsylvania, PA 16802, USA. [3] Institute for CyberScience, The Pennsylvania State University, Pennsylvania, PA 16802, USA. [4] Ernst Strüngmann Institute (ESI) for Neuroscience in Cooperation with Max Planck Society, Frankfurt am Main D-60528, Germany. [5] Neurophysiology Imaging Facility, National Institute of Mental Health, National Institute of Neurological Disorders and Stroke, and National Eye Institute, National Institutes of Health, Bethesda, MD 20892, USA. [6] Section on Cognitive Neurophysiology and Imaging, Laboratory of Neuropsychology, National Institute of Mental Health, National Institutes of Health, Bethesda, MD 20892, USA. Correspondence and requests for materials should be addressed to X.L. (email: xxl213@engr.psu.edu)

Rapid progress is being made towards revealing the human brain's specialized functional architecture using fMRI studies of spontaneous activity. In the absence of behavioral tasks, spontaneous brain activity fluctuates in an organized manner, with its correlational structures exhibiting a topology resembling that of brain networks known to support overt behavior[1,2]. This capacity to tap into functional networks during rest has opened the door to the study of brain circuits that are difficult to probe by behavioral tasks, and is a promising avenue for the study of brain pathologies that affect these networks[3]. Recent initiatives like the UK Biobank[4] and the Human Connectome Project (HCP)[5] are using resting-state fMRI to develop normative databases to facilitate this research.

Despite this progress, a major confound in resting-state fMRI —that may limit widespread clinical application—is the phenomenon of brain-wide (often called global) fluctuations in activity that is often present during these studies. Although still poorly understood, this global activity may result from changes in brain state, possibly mediated by an afferent neural signal that is common across cortical regions. It can lead to fMRI signal correlations suggestive of functional connections where there are none. There is converging evidence that fluctuations in vigilance and arousal contribute to this global activity. For example, electroencephalography (EEG) alpha activity, an indicator of vigilance, generally shows widespread negative correlations with the fMRI signal[6–9] and this has also been demonstrated for cortical arousal[10–12]. Consistent with this, caffeine administration has been shown to reduce this global fMRI activity[13], whereas the opposite effect has been observed with several hypnotic drugs, including midazolam and zolpidem[14–16]. Transient, widespread fMRI activity has also been observed during behavioral tasks, which may also be attributable to momentary changes in vigilance[17]. In light of these findings, a proper interpretation of resting-state fMRI in terms of functional network topology will require a precise knowledge of—and the ability to account for— the contribution of arousal to the fMRI signal.

A recent study applying spectral analysis to electrocorticography (ECoG) data from macaques suggest that decreases in arousal may be at the root of widespread transient and synchronous cortical activity[18]. During rest, the ECoG signal was marked prominently by a series of sequential spectral transitions (SSTs), whose stereotyped pattern elapsed over several seconds and consisted of a decrease in mid-frequency activity, accompanied by an increase in gamma-band activity and followed by an increase in delta-band activity. These SSTs involve large swaths of cortex, consistent with their potential role in the global fMRI signal. Nevertheless, the hypothesized involvement of SSTs in the global fMRI signal requires further confirmation, which is complicated by the difficulty in concurrently acquiring fMRI and electrophysiological data. Moreover, no association between the global fMRI signal with the neural circuits known to regulate brain arousal has yet been demonstrated.

In the present study, we investigated the neural origin of the global fMRI signal and its association with arousal. To do so, we capitalized on the brain coverage afforded by fMRI, by first linking the electrophysiological SST events with fMRI motifs in macaque monkeys, and then using this information to probe a large data set of high-resolution human fMRI data from the publicly available HCP database. Using this approach, we demonstrate a close association between the global fMRI signal and SSTs, and furthermore establish a subcortical involvement in both features that is consistent with an origin in arousal. We discuss these findings in light of a possible regulation of cortical excitability by long range projections emanating from the basal forebrain, as well as the implications of this global activity for human resting-state fMRI studies.

## Results

**Electrophysiological correlate of the global fMRI peaks.** To determine the fMRI signature of electrophysiological SSTs, we reanalyzed fMRI and LFP data that were concurrently acquired in macaques and have been published previously[19]. The LFP data, collected from one or two brain sites in each of three macaques (monkeys A, S, and V) were converted to mean spectrograms, which facilitated the identification of SSTs. The global fMRI signal of each monkey was extracted by averaging across the brain. Details of these procedures are given in the "Monkey data" subsection of the Methods section.

The SST patterns were clearly visible in the mean spectrograms (Fig. 1a, top) and closely resembled those observed previously in large-scale ECoG recordings (Fig. 1b)[18]: momentary reductions in mid-frequency (10–30 Hz) power, closely followed by increases in slow-wave (<4 Hz) and gamma-band (>40 Hz) power. We furthermore noticed that SSTs were followed by a large, widespread increase in fMRI signal, which appears as large peaks in the global signal (Fig. 1a, bottom). Notably, $90.64\% \pm 3.10\%$ (mean ± SD across $N = 16$ sessions) of brain voxels show signal increase at the top 50% global signal peak points. To investigate this apparent SST–fMRI relationship more quantitatively, we averaged mean spectrograms in 40-s segments centered on the top 50% global signal peak points, i.e., local maximums ($N = 125$, 401, and 197 for monkey A, S, and V, respectively) (Fig. 1a). The resulting average spectrogram showed a clear SST pattern (Fig. 1c), which closely resembles the one derived previously from ECoG data (Fig. 1b) (the peak correlation at the delay of 5.2 s: $0.46 \pm 0.08$ across $N = 16$ sessions, $p = 7.3 \times 10^{-13}$, one-sample $t$-test). In contrast, the average spectrogram with respect to time-reversed control points failed to show the same pattern (the correlation at the delay of 5.2 s: $0.006 \pm 0.123$ across $N = 16$ sessions, $p = 0.84$, one-sample $t$-test) (Fig. 1d). Consistent results were obtained across different monkeys and over individual sessions from the same monkey (Supplementary Figure 1).

**Co-activation pattern of global fMRI signal increases.** Having established a link in the monkey data between the global fMRI signal and SSTs, which have been attributed to arousal changes, we asked whether these arousal-related SST events might contribute to the resting-state fMRI signal in humans. To this end, we analyzed human resting-state fMRI data from a subgroup of subjects from the HCP database[5]. We first analyzed resting-state fMRI data from a subgroup of 94 (~20% of 469) subjects, who were selected because their overall global (i.e., averaged over gray matter) resting-states signal showed the largest temporal variation. The four runs were collected in pairs on different days to allow for an assessment of reproducibility. On average, the global signal variation explained a substantial proportion ($15.32\% \pm 3.69\%$) of the total signal variance seen in individual gray matter voxels.

To determine which neural structures may contribute to large global fMRI signal increases during rest, we first examined the spatial distribution of the global fMRI peaks in a manner analogous to a dynamic approach that temporally decomposes correlated network activity into co-activation patterns[20,21]. Briefly, the fMRI signals were standardized to reflect changes, i.e., co-(de)activations, relative to their temporal mean, and then the co-(de)activations at time points with a global fMRI signal larger than a pre-defined threshold of 0.2, corresponding to the top 16.6% time points, were averaged to generate a co-activation pattern for the global signal increase (Fig. 2a). We found the co-activation pattern involved significant ($p < 0.01$, $N = 18,725$ time points, permutation test, no multiple comparison correction)

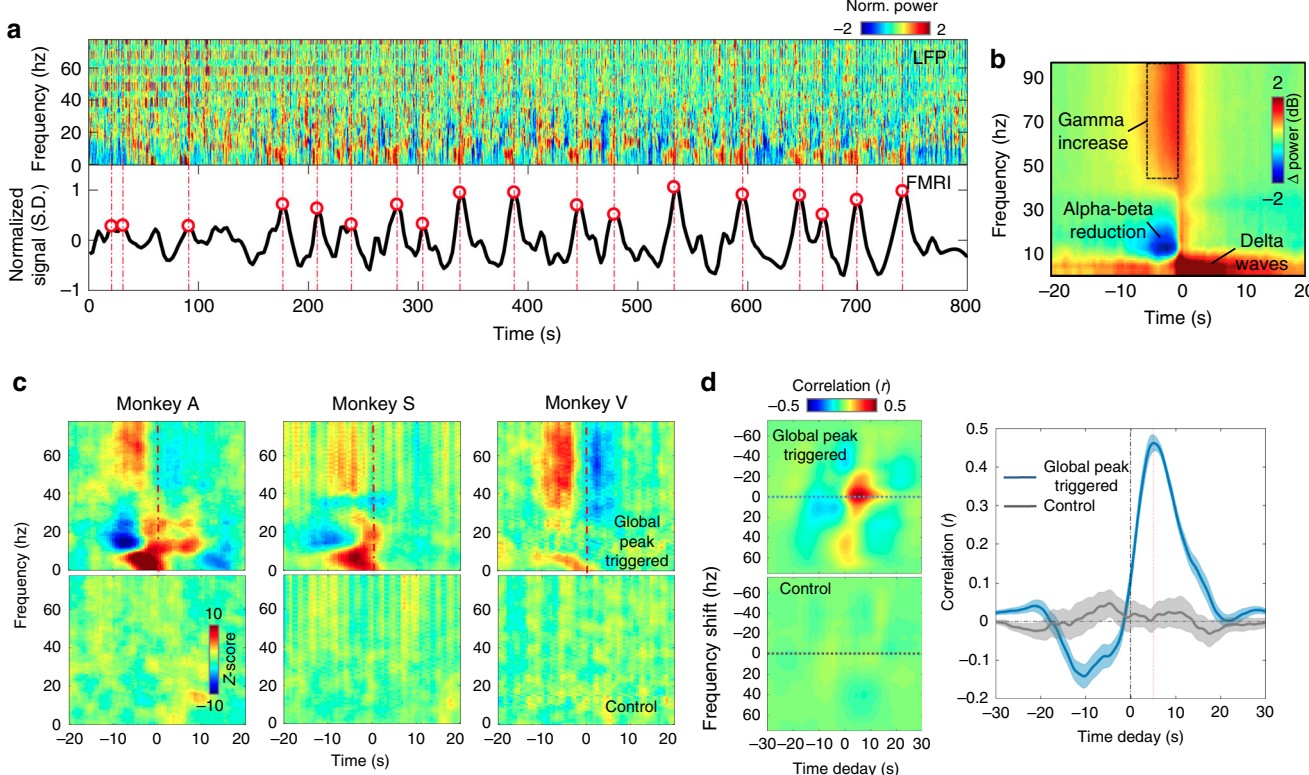

**Fig. 1** Global fMRI peaks and electrophysiological SST events in monkeys. **a** The large peaks in the global fMRI signal (the lower panel) are preceded by SST events in the mean spectrogram of concurrently acquired LFP signals (the upper panel). **b** The SST pattern was obtained on an independent ECoG data set from macaque monkeys by a previous study[18]. **c** The SST pattern emerges with aligning and averaging the mean LFP spectrogram segments with respect to the top half of large global fMRI peak points (marked as red circles and dash lines in the example in **a**). In contrast, the control cases, which were obtained by reversing time locations of these peak points, do not show any obvious pattern. The results for monkey A, V, and S are averaged from 125, 401, and 197 data segments, respectively. **d** The 2D cross-correlation function between the ECoG SST pattern (**b**) and the global peak triggered LFP pattern shows a clear peak (top left, averaged across N = 16 sessions), and its cross-sectional profile at the zero-frequency shift reaches the maximum at a time lag of 5.2 s, as expected based on the fMRI hemodynamic delay (the blue curve in the right panel, the shadow represents regions within one S.E.M.). In contrast, the 2D cross-correlation function (bottom left) and the corresponding zero-frequency profile (the gray curve in the right panel) for the control case do not show clear peaks

signal increase in 93.9% of gray matter voxels (colored regions in Fig. 2b). Notably, the strongest activity was seen in the early sensory cortices (Fig. 2b), similar to the distribution of gamma-band spectral power increase at SSTs seen in monkey ECoG data (Fig. 2c). Specifically, co-activation amplitudes in the primary somatosensory (0.41 ± 0.25 across N = 5216 voxels), motor (0.40 ± 0.23 across N = 2452 voxels), auditory (0.39 ± 0.18 across N = 2216 voxels), V1 (0.41 ± 0.22 across N = 5142 voxels), and V2 visual (0.42 ± 0.19 across N = 3568 voxels) cortices are significantly higher (p = 0 for all comparisons, two-sample t-tests) than other gray matter regions (0.12 ± 0.17 across N = 320,040 voxels). The similarity in spatial distribution provided the first suggestion that the gamma-power bursts at SSTs, attributed to momentary increases in cortical excitability concurring with arousal drops, underlie positive-going peaks in the spontaneous fMRI signal. This relationship is consistent with previous reports of a tight coupling between gamma-band power and fMRI signal[22,23].

Further inspection of the areas within the HCP data set that were co-(de)activated with the global fMRI peaks revealed three subcortical regions in the basal forebrain, thalamus, and midbrain, respectively, showing decreased signals during positive fMRI peaks. The most conspicuous decreases (147 voxels with Z < −6, the average Z-score: −9.07 ± 2.58, the peak Z-score: −16.57) were found in a flat, horizontal structure in the transverse plane spanning only ~2 fMRI voxels (4 mm) in the superior-inferior direction and situated between the anterior commissure and optic

tracts (Fig. 3a). The location of this structure corresponded well to the anatomical location of the nucleus basalis (NB) of the basal forebrain (Fig. 3b), a major node of the central arousal system[24,25]. On the basis of a comparison with a stereotaxic cytoarchitectonic map of the human basal forebrain, this region of decreased activity (Z < −6) was found to largely overlap (65.3% of the 147 voxels) over the established location of Ch4 cholinergic neurons, dominant cell types of the NB[26,27] (Fig. 3d). Further indication that this area of activity decreases corresponded to NB came from its shape, which was found to be consistent with the distribution of Ch4/Ch4p cholinergic cells: its anterior portion bends dorsally and it transitions sharply from a more anterior, medial region (Ch4) at the inferior slice (Z = −11) to a more posterior, lateral region (Ch4p) at the superior slice (Z = −9) (Fig. 3c). Significant, but smaller (in amplitude and/or spatial extent) decreases were also seen in two other subcortical regions: a small patch at the dorsal midline thalamus (23 voxels with Z < −6, the average Z-score: −11.91 ± 4.24, the peak Z-score: −20.33) (Fig. 4a), likely covering the paraventricular nucleus (PVT) and the medial portion of magnocellular mediodorsal nucleus (MD), and a symmetric midbrain structure just above the Pons (11 voxels with Z < −6, the average Z-score: −8.25 ± 1.02, the peak Z-score: −9.55) (Fig. 4b), which we tentatively regarded as the substantia nigra (SN) based on comparison with the atlas. Of note, fMRI co-activation patterns derived from the monkey fMRI data showed a similar spatial distribution (Supplementary

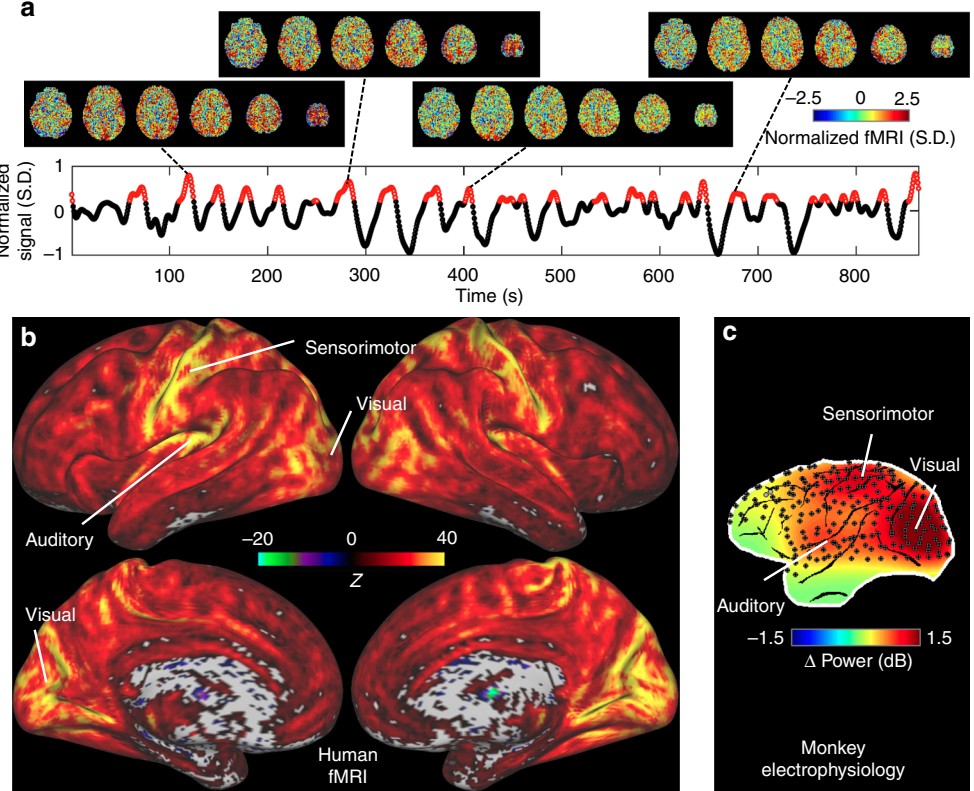

**Fig. 2** Global fMRI co-activation pattern in a representative human subject. **a** A global fMRI time course from a typical subject and four examples of instantaneous co-activation patterns at individual time points with large global signal. The time-averaged global co-activation pattern was generated by averaging all the high-signal (red) time points, which were defined to have a global signal larger than 0.2 and made up 16.6% of total number of time points. **b** The human fMRI co-activation pattern at time points with high global signal. The Z-score map suggests that the visual, auditory, and sensorimotor regions show much larger and more consistent signal increase than other brain areas at time points with large global signals, even though the 93.9% gray matter voxels (colored regions) show a statistically significant score ($Z > 2.58$, equivalent to $p < 0.01$, $N = 18,725$ time points). **c** The gamma power increases at the SST (within the dash box in Fig. 1b) seen in monkey electrophysiology show larger amplitude at the visual, auditory, and sensorimotor regions. The results here are adapted from ref. [18]

Figure 2), although the poorer spatial resolution prevented a detailed comparison.

Lastly, we investigated whether the temporal scale of co-activations in the HCP data was consistent with that of the SST. For that purpose, we averaged the fMRI signal around the occurrence of global peak points ($N = 2134$) at time lags between −15 and 15 s. It was observed that co-activations spanned about 10 s, consistent with the approximate duration of an SST (Fig. 5).

It should be noted that the findings reported above were obtained from only 20% of the subjects showing the largest global signal fluctuations, and thus may not be typical of the general population. This concern is alleviated, however, by the fact that the findings were reproduced by analyzing the data across two sessions from two different days, from the same number of randomly selected subjects ($N = 94$), and from a much smaller number of subjects ($N = 20$) (Fig. 6). Another potential concern is that the apparent subcortical de-activations were an accidental result, due to weak fMRI signals in these subcortical regions, which themselves have been associated with magnetic field inhomogeneity. However, as the absolute fMRI signal, i.e., the temporal mean, was removed from the analysis during the de-meaning processing and the co-(de)activations quantify changes relative to this mean, there should not be any causal relationship between the two. In addition, the pattern of the global co-(de) activation and mean fMRI signal strength appear to have little in common (Supplementary Figure 3), consistent with this theoretical prediction.

## Discussion

Here we elucidated the electrophysiological underpinnings of global signal variations widely observed in resting-state fMRI data. We first demonstrated a link between momentary increases in the global fMRI signal and SSTs in LFP data. We then analyzed high-resolution HCP fMRI data, and found evidence for a link to arousal drops by showing that momentary increases in the global signal are accompanied by signal decreases in subcortical regions with established involvement in promoting arousal and wakefulness.

The specific pattern of subcortical involvement included the NB and, to a lesser extent, the dorsal midline thalamus region and the midbrain region tentatively ascribed to the substantia nigra. Altogether, these regions are well established nodes in arousal pathways influencing the telencephalon[28]. Lesions to the NB can result in increased delta power in cortical field potentials[24], which is considered a signature of low vigilance[29] and emerges at the later part of an SST. EEG-fMRI studies have shown that the fMRI signal in the dorsal midline thalamus positively correlates with alpha-band EEG power, an indicator of vigilance, and negatively with primary sensory cortices[7,9]. The midline region of the thalamus, along with the intralaminar nuclei, are often considered a part of the non-specific arousal system in the brain, and it has been shown recently that optogenetic stimulation of midline thalamus neurons effectuates arousal in naturally sleeping mice[30]. Furthermore, damage to the midline thalamus in humans can result in a conversion to a vegetative state[31]. Altogether, these

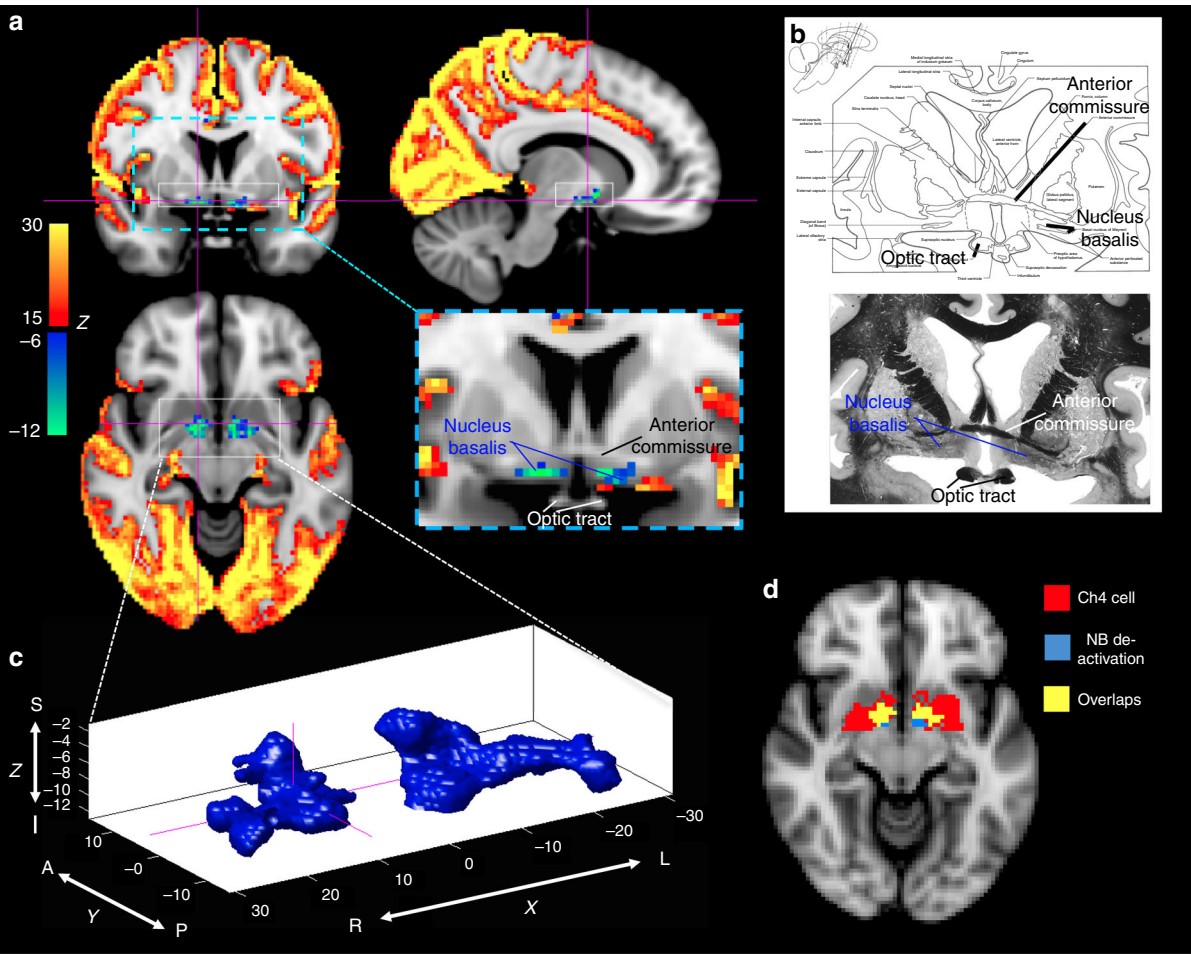

**Fig. 3** De-activation of nucleus basalis (NB) during the global fMRI co-activation. **a** The global signal co-activation map is overlaid on a high-resolution anatomical MRI scan and shown in three slices at sagittal ($X = 11$), coronal ($Y = 2$), and axial ($Z = −10$) planes, respectively. All coordinates are in the standard Montreal Neurological Institute (MNI) space, which offsets in $Y$ and $Z$ directions by 4 and 5 mm with respect to the "anatomical MNI space" previously used for the basal forebrain cytoarchitectonic mapping[26]. **b** The de-activated region from coronal view is amplified and compared with the NB region shown in a human brain atlas[69]. **c** 3D rendering of the de-activated regions shown in **a**. **d** The overlap (yellow) between the NB de-activation mask (light blue, corresponding to $Z < −6$) and a mask of Ch4 cell distribution (red)[27] derived from the cytoarchitectonic map of the human basal forebrain[26] is shown at an axial slice ($Z = −10$)

findings suggest the important role of the midline thalamus in promoting and maintaining arousal and wakefulness. The negative patch at the midline thalamus spans about 4–6 mm (2–3 voxels) along the left-right axis and thus covers both the PVT (about ~1 mm in each hemisphere in the same direction) and the medial portion of the magnocellular MD. It is possible that only one of these two thalamic nuclei contributes to the observed fMRI signal decrease, but this cannot be determined from the limited (2 mm) resolution of the fMRI data analyzed here. However, it is worth noting that both nuclei receive inputs from arousal-promoting nuclei in the basal forebrain, hypothalamus, and brainstem, including the tuberomammillary nucleus, locus coeruleus, and dorsal raphe nucleus. In particular, the MD receives cholinergic innervations from the laterodorsal tegmental nucleus, the pedunculopontine nucleus, as well as the basal forebrain, with an uneven distribution dominated in the medial portion of MD[32,33]. These anatomical connections provide a structural basis for the relevance of the midline thalamus region to arousal regulation and may also explain its co-deactivation with the NB region observed in this study. SN lesions in rats have also been shown to affect various aspects of sleep, including a decrease in the latency to the onset of slow-wave sleep[34]. Therefore, all three subcortical sites have been shown to have an important role in

promoting wakefulness, and their de-activation is consistent with a reduction in arousal level, which is surprisingly associated with a widespread increase in cortical fMRI signals as we observed herein.

Although the observed reduction in fMRI signal in the arousal-promoting basal forebrain regions is consistent with the notion of arousal drops, the increase in global fMRI signal at the cortical level may seem counter-intuitive, as states of low arousal (e.g., sleep) are commonly associated with low levels of cortical activity. Likewise, the divergent changes in the cortical and subcortical regions also seem inconsistent with the observation that stimulating the NB caused cortical excitation reflected as an increase in gamma-band EEG power[35]. However, it is possible that the arousal drops leading to prominent fMRI signal increases are unique to state transitions in which alertness and awareness are fading away. The mid-frequency alpha-beta-band brain activity has been hypothesized to reflect a top-down, feedback inhibitory process unique to alert and awake conditions[36–38]. The mid-frequency activity reduction at the early phase of the SST may signify a fade-away of this inhibitory process at the transition of consciousness states, which momentarily disinhibits gamma-band activity and facilitates an fMRI signal increase. The fMRI signal increases may thus be specific to lapses in alertness secondary to

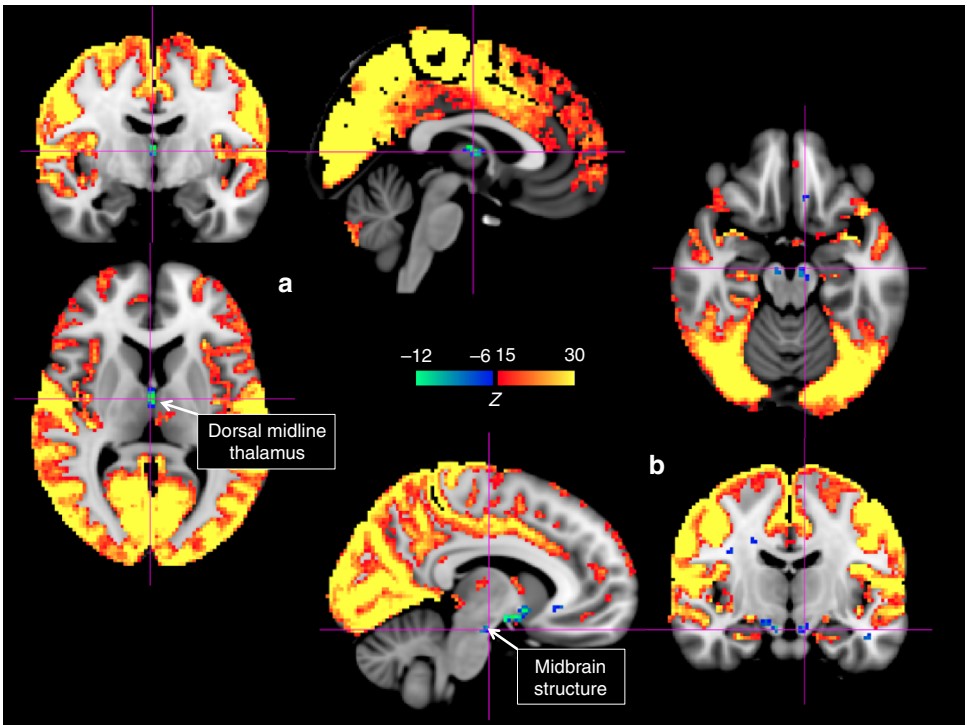

**Fig. 4** De-activation of two other subcortical regions during global signal co-activation. The de-activation in the thalamus is situated at the dorsal midline region, as shown in three imaging slices at sagittal ($X = 2$), coronal ($Y = -8$), and axial ($Z = 7$) planes, respectively. **b** The de-activation in the midbrain is located right above the Pons, as shown in three imaging slices at sagittal ($X = -8$), coronal ($Y = -14$), and axial ($Z = -18$) planes, respectively. This midbrain structure is tentatively assigned to be the substantia nigra (SN) based on a comparison with the human brain atlas[69]

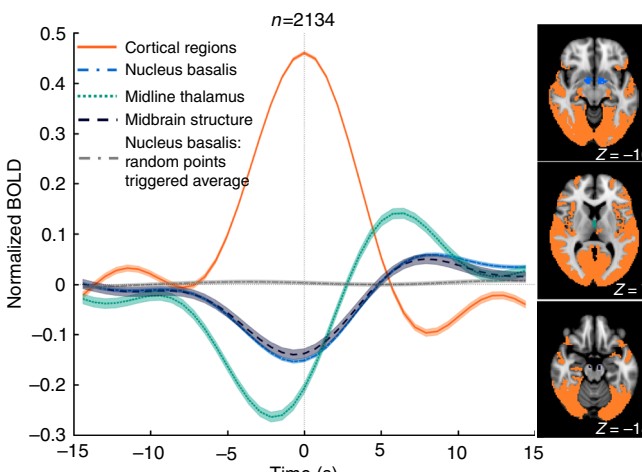

**Fig. 5** Temporal dynamics at global fMRI signal peaks. The averages of fMRI signals around the global signal peak points ($N = 2134$) at the nucleus basalis of the basal forebrain (blue, dash-dot line), the dorsal midline thalamus (green, dotted line), the midbrain structures (dark blue, dashed line), and the cortical region (orange, solid line). The control (gray, dash-dot line) is an average of fMRI signals around randomly selected time points at the nucleus basalis. The shadow regions represent the area within one standard error of the mean. The masks for de-activations at the nucleus basalis (blue), the dorsal midline thalamus (green), and the midbrain structure (dark blue) are shown at three axial slices at $Z = -10$, 7, and $-18$, respectively, along with the mask of the cortical co-activations (orange) on the right panel

—or associated with—a transition to a state of lower arousal. This is also consistent with the SST-like feature observed at the induction of propofol anesthesia: although the initial arousal reduction from high arousal level is associated with a transient surge of gamma-band ECoG power, this is followed by a subsequent drop to below-baseline levels, presumably associated with the low arousal of unconsciousness[18]. The potential role of the mid-frequency activity in mediating top-down inhibition also raises the possibility that the cortical changes in these arousal events precede the subcortical modulations, especially given that that mid-frequency power reduction has been shown to be stronger in brain regions that project to the subcortical nodes of the ascending arousal system[18,39]. However, it should be emphasized that the presented analysis is purely based on correlations and does not permit one to infer a causal relationship between cortical and subcortical activity; this is a challenge that remains to be addressed in future studies. The NB–cortex interactions under different brain states could be mediated by different groups of neurons given the finding that GABAergic neurons in the basal forebrain are more involved in its wake-promoting function, whereas cholinergic cells are more closely related to its modulatory effects on cognitive functions during wake[40].

We found that arousal modulations are associated with dichotomous fMRI changes in widespread cortical areas and subcortical wake-promoting regions, including the NB of the basal forebrain. Although a similar cortical-subcortical dichotomy has been reported previously in fMRI studies of arousal, they focused on the thalamus and the brain conditions with task performance or pathological brain activities. For example, a previous neuroimaging study found a specific role of the thalamus in mediating the interaction of selective attention and arousal[41]. Furthermore, concurrent fMRI-EEG recordings in human epilepsy have shown that absence seizures accompanied by generalized spike-wave discharge activity, are associated with

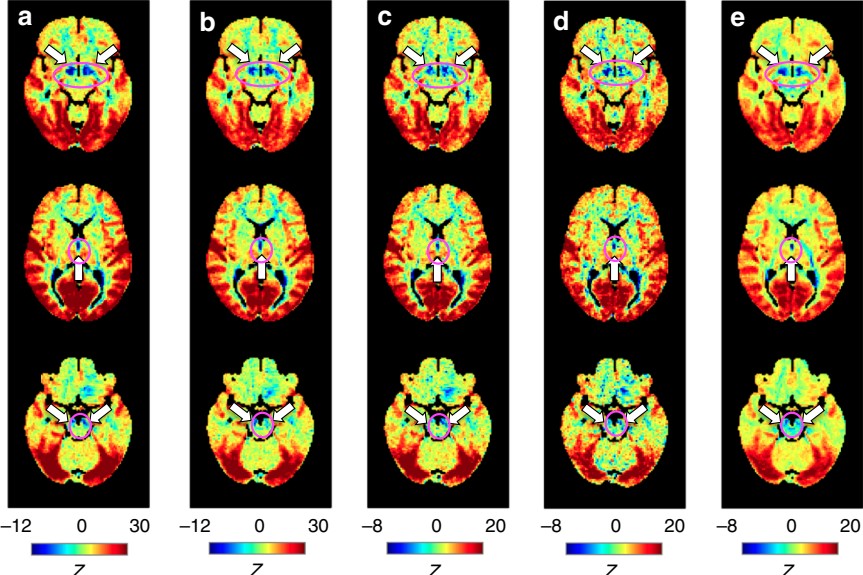

**Fig. 6** Reproducibility of the global signal co-activation pattern. Global signal co-activation maps derived from the DAY 1 (**a**) and DAY2 (**b**) data of the top 94 subjects with the largest global signal fluctuations (i.e., standard deviation) in DAY 1 data, from the DAY 1 data of 94 randomly selected subjects (**c**), from the DAY 1 data of top 20 subjects showing the largest global signal fluctuations (**d**), as well as from the DAY 1 data of top 94 subjects but without using ICA-FIX preprocessing (**e**). The result in **a** is the same as that shown in Fig. 3. Three slices are at $Z = -10$, 7, and −18, respectively. The white arrows and magenta circles in the top, middle, and bottom rows indicate de-activations in the nucleus basalis (NB) of the basal forebrain, the dorsal midline thalamus, and the midbrain structure, respectively

widespread fMRI signal decreases in the cortex and focal increases in the median portion of the thalamus[42]. Although this anti-polar cortical–subcortical relationship was also seen in our study, it is not clear how the difference in the direction of changes between the studies can be reconciled. It is possible that the sustained (up to 1 min) low-frequency (~3 Hz) generalized spike-wave discharges is similar in nature to neural activity during the late stage of the SST, where fMRI signal polarity reverses and becomes consistent with that seen in epilepsy.

In animals, anesthetic conditions characterized by a burst-suppression EEG pattern have also shown widespread fMRI signal fluctuations. These co-occur with EEG bursts that either present spontaneously[43] or are elicited by sensory stimulations[44–46], and appear somewhat stronger at sensory regions[45,46]. It is, however, unclear whether the EEG bursts during anesthesia and the SSTs in the drowsy state are generated by similar mechanisms. In this regard, it should be realized that the coupling between the local electrophysiological signal, neuronal energy consumption, and the fMRI response is complex and may vary across arousal states[46].

The findings reported here have important implications for fMRI studies of brain functional connectivity. The global resting-state fMRI signal introduces spatially non-specific signal correlations that lead to overestimation of local functional brain connectivity[47,48]. For example, studies of non-stationary aspects of network activity[20,49], disease-related network modulations[50,51], and behavioral correlation of network activity[52], could be susceptible to fluctuations in arousal and vigilance. This is particularly problematic when studying patient populations with potentially abnormal arousal regulation, including Alzheimer's disease[53], Parkinson's disease[54], and autism spectrum disorders[55]. This concern is quite realistic in the light of a previous meta-analysis of resting-state data that found that a third of subjects likely fell asleep within 3 min[12], which is at the low end of durations for typical resting-state fMRI experiments.

Given a strong influence of arousal state on the fMRI signal, and the difficulty in minimizing arousal variations, the most fruitful way to reduce confounding effects on the interpretation of resting-state data may be to remove or account for them in post-processing. A simple strategy is to remove the global signal component, as it is clear that this signal is closely related to arousal fluctuations. There are multiple approaches proposed to achieve this, for example, by regressing out the global average or taking out the first principal component of resting-state fMRI signals[56–58]. Although this strategy has been controversial in whether it will introduce spurious negative fMRI correlations[59,60], some consensus has been developed recently regarding its benefits and disadvantages[48]. Caution should also be exercised when applying the global signal regression method to resting-state fMRI data sets without much global signal changes, since it can potentially remove other types of fluctuations not related to arousal. An alternative strategy would be to identify and then exclude time points showing spatio-temporal patterns similar to either that of global co-activations we observed in this study or to the one derived based on fMRI correlations with an EEG-based arousal index[10]. This second strategy may be most specific to arousal effects and have less unintended effects on the data.

To summarize, this study demonstrated a tight coupling between a temporally distinct electrophysiological event, the SST, and widespread fMRI activity whose spatial and temporal characteristics are consistent with an origin in momentary drops in arousal. The presented findings clarify the electrophysiological correlate for the global fMRI signal and elucidate its relationship with arousal, paving the way for methods that properly account for arousal fluctuations during resting-state fMRI.

## Methods

**Monkey data.** The concurrent fMRI-electrophysiology data have been reported in a previous publication[19], which includes more details about the data set. To summarize here, all procedures followed National Institutes of Health Guidelines and were approved by the Animal Care and Use Committee. The fMRI and electrophysiological signals were concurrently recorded from three female rhesus macaque monkeys (A, S, and V) in a 4.7 T/60 cm vertical scanner (Bruker BioSpec 47/60). The monkeys had been acclimated to the MR environment and trained to sit quietly in the scanner. The data were acquired when the monkeys sat in near-

complete darkness without a monitor screen being present and with an infrared video camera sitting inside the bore to monitor and record the animal's behavioral state. The three monkeys had different behavioral tendencies: monkey A had a strong tendency to close her eyes (possibly falling asleep), monkey S appeared to have interleaved periods of sleep and wakefulness, and monkey V typically remained alert throughout the session.

Before each experiment, the monkeys were intravenously administered a dose (8–10 mg kg$^{-1}$) of monocrystalline iron oxide nanoparticles (MION), which is a MR contrast agent that can isolate regional cerebral blood volume (rCBV) component and yield an improved signal-to-noise ratio compared with the blood oxygenation level-dependent contrast. A gradient-echo echo-planar imaging sequence was used to acquire a total of 700 fMRI volumes for each run with a repetition time (TR) of 2.6 s, which consists of a 1.6-s acquisition and a 1.0-s gap. There are 15 and 16 axial slices for monkey A and S and 19 coronal slices for monkey V with a thickness of 1.5 mm, an inter-slice gap of 0.5 mm, and an in-plane resolution of 1.25 mm × 1.25 mm. We analyzed a subset of the previous data set that includes 3, 9, and 4 runs from monkeys A, S, and V, respectively. As functional activations lead to increased rCBV that manifests a decrease in MION-based fMRI signal, we invert the sign of fMRI signal throughout the paper to avoid confusion.

MR-compatible chronic electrodes (NeuroNexus Technologies) were used for electrophysiological recordings. For the part of the data set we analyzed, two linear electrode arrays of 16 contacts with a diameter of 40 µm and a spacing of 100 µm between contacts were inserted into the parietal cortex (area 7a) and frontal cortex (area 6d), respectively, in monkey A. A linear electrode array of 32 contacts was implanted in the primary visual cortex (V1) of monkey V. A subdural electrocorticography (ECoG) array of 32 contacts that was arranged in a regular grid (6 × 6 minus the corners) with 600-µm spacing was placed in the area V4 of monkey S. LFP recordings were conducted inside the scanner bore with a sampling rate of 5 kHz using an MR-compatible 32-channel amplifier with an input dynamic range of 16 mV and a 16-bit A/D converter (BrainAmp; Brain Products). The signal was band-pass filtered between 0.5 Hz and 1000 Hz. The scanner was synchronized to the electrophysiological data via a brief digital pulse delivered at the beginning of each volume acquisition.

**HCP data.** We used the HCP 500-subject data release, which includes 526 subjects. All data were collected on a 3 T Siemens Skyra scanner with gradients customized for the HCP. We restricted our analysis to 469 subjects (age = 29.2 ± 3.5, range: 22–36, 275 females) who have all four sessions of resting-state fMRI in full length. The resting-state fMRI data were acquired in four separate 15-min runs on two different days (two per day) with a 2-mm isotropic spatial resolution and a 0.73-s temporal resolution. This excellent quality superior to most neuroimaging studies was made possible by using the multiband echo-planar imaging with an acceleration factor of 8[61].

**Preprocessing.** The fMRI signals from monkeys were linearly de-trended and then band-pass filtered between 0.005 Hz and 0.1 Hz. The resulting fMRI time series were standardized by subtracting the temporal mean and then dividing by the temporal standard deviation. For the LFP data, the first 1.75-s segment in each fMRI volume acquisition, i.e., the 1.6-s segment with fMRI data acquisition and the first 0.15 s during the gap period, was discarded to remove signals contaminated with MR-gradient induced artifacts and the remaining 0.85 s data segments were concatenated for further analysis. To see the SST events at time-frequency domain, the LFP signal of each channel was time-frequency transformed (1 s window, 0.2 s sliding step, and 5 tapers) to a spectrogram using a multi-taper method implemented in Chronux, a matlab package for analyzing the neural data[62]. The resulting spectrogram was converted to a decibel (dB) unit using a logarithmic operation and shows the temporal evolution of LFP power at each 1-Hz frequency bin between 1 and 100 Hz. To equalize spectral power and fluctuation amplitude at different frequencies, the power fluctuation at each frequency bin was further normalized by subtracting the temporal mean and dividing by temporal standard deviation, similar to the normalization of fMRI signals. These processing steps are identical to those used in the previous monkey ECoG study of SSTs[18]. As 67.3% LFP signals were discarded at every 2.6 s, the spectrogram was rescaled in time to match the timing of the fMRI signals.

The HCP data were preprocessed using the HCP MR minimal preprocessing pipeline[63], which combines a set of tools from FSL, FreeSurfer, and the HCP Connectome Workbench[64]. It includes three MR structural pipelines for distortion-correction using a $B_0$ field map and alignment of individual brains to a common MNI template with the use of high-resolution (0.7-mm isotropic) $T_1$-weighted and $T_2$-weighted MR scans. For the 500 subjects release, individual surface registration was achieved by combining the cortical folding surface features and multimodal surface matching algorithm[65]. The fMRI data were preprocessed with the MR functional pipelines, which include distortion-correction, correction for subject motion by realigning fMRI volumes, registering the fMRI data to the structural data, and also converting fMRI time series from the volume into the gray-ordinates standard space. After the minimal preprocessing pipeline, the resting-state fMRI data were further denoised using the novel ICA-FIX method[66]. We conducted parallel analyses on data with and without using ICA-FIX

denoising, and obtained very similar results. The main results reported in this paper are from the ICA-FIX denoised data only.

In addition to the preprocessing steps implemented by the HCP, we applied mild smoothing both spatially (Gaussian filter with the FWHM = 2.4 mm) and temporally (band-pass filtered at 0.005–0.1 Hz). We then standardized each voxel's signal by subtracting its temporal mean and dividing by its temporal standard deviation. This normalizing step equalized variance of fMRI time series and thus eliminated potential biases caused by varying fMRI fluctuation amplitude across brain regions. Unfortunately, as a drawback, it prevented absolute quantification in terms of fractional fMRI signal changes.

**Global fMRI peak triggered averaging of LFP data from monkey.** The normalized fMRI signals were averaged with a mask covering brain tissues to generate a global fMRI signal. Peak points, i.e., local maxima, in the global fMRI signal were identified, and those with the largest amplitude (top 50%) were selected. The LFP spectrograms were also averaged across all channels to generate a mean spectrogram. After that, 40-s segments of this mean spectrogram, centered on these selected peak points, were aligned and then averaged to create an averaged time-frequency pattern around the large global fMRI peaks. To produce a control for this analysis, the identified fMRI peak point locations were reversed in time for each run; for example, the peak point at the 28th volume was placed at the 673th volume (28 volumes away from the last volume, as the total length of a run was 700 volumes). An identical averaging procedure was performed with respect to these false peak positions in the control case. Then, the resulting averaged spectrogram segments were converted to Z-scores according to the distribution of its values in the control case.

To assess the similarity between the global peak triggered average of LFP spectrogram segments and the SST pattern derived previously in an independent ECoG data set, their 2D cross-correlation function was calculated with frequency shifts between ±75 Hz and time shifts between ±30 s. The 1-dimensional cross-correlation function was also extracted at the zero-frequency shift, and the peak correlation was identified and whether it is statistically different from zero was assessed using a one-sample t-test. The same procedure was repeated on the control cases of the time-reversed peak point locations.

To estimate the spatial extent of fMRI signal increases, fMRI volumes at these top 50% peak points were averaged for each session, and the percentage of brain voxels showing a signal increase relative to the temporal mean, which is equal to zero, were calculated and summarized.

**Global signal co-activation pattern in HCP data from human.** To minimize the contribution from non-neuronal sources, the global signal was calculated for each subject by averaging the standardized fMRI signals within the gray matter. We then used the standard deviation of the global signal to quantify its fluctuation amplitude, and selected 94 (~20% of 469) subjects (age = 28.8 ± 3.6, range: 22–35, 37 females) showing the largest fluctuation amplitude for subsequent analyses. We adapted a method previously used to temporally decompose a single resting-state network into multiple co-activation patterns to create a co-activation map corresponding to global signal peak[20]. Briefly, time points with a global signal value larger than 0.2 (equivalent to the top 16.6% time points) were selected, and the fMRI volumes at these time points were simply averaged to generate the global signal co-activation map. We then compared this resulting map to a null distribution built by simply averaging fMRI volumes at groups of randomly selected time points to convert it into a Z-score map. Significant negative Z-scores were widely observed in sulcal and ventricular cerebrospinal fluid, especially near the large pial and ependymal veins, which is similar to the finding in a previous resting-state study and may be attributed to the cerebral blood volume (CBV) effect in large veins[67]. To focus on the de-activation in neural tissues, we applied a mask of the cerebrospinal fluid regions to mask out results from non-tissue regions.

To assess the spatial heterogeneity of the global signal co-activation amplitude, we defined five regions of interest based on the Juelich histological atlas[68]: the primary somatosensory cortex, the primary motor cortex, the primary auditory cortex, the V1 and V2 cortices. Before the global signal co-activation map was converted to the Z-score map, we summarized the map values within these five regions of interest, and compared their averages with the mean of all other gray matter voxels using two-sample t-test.

To obtain the dynamic information about the cortical co-activations and subcortical de-activations at global signal peaks, four masks were generated based on the global signal co-activation map to cover regions showing large cortical co-activations (Z > 15) and de-activations (Z < −6) in the NB, dorsal midline thalamus, and midbrain structure, respectively. The fMRI signals within these four ROIs were then aligned to and averaged at the 2134 global signal peak points, i.e., local maxima, over the threshold (i.e., 0.2), as well as at the randomly selected time points (N = 2134) to serve as controls.

To assess the location of the NB de-activation relative to that of basal forebrain Ch4 neurons, we calculated the percentage overlap of the NB de-activation mask derived above with a previously defined mask of the Ch4 neurons[27] based on a stereotaxic cytoarchitectonic map of the human basal forebrain[26].

As some subcortical areas are known to suffer significant fMRI signal loss due to magnetic susceptibility artifacts, we examined whether subcortical de-activations in the global signal co-activation pattern were related to abnormally low fMRI

signals in corresponding regions. To do this, fMRI signals of each subject were first normalized with respect to their grand means averaged over time and brain voxels, and the 112,800 fMRI volumes from the 94 subjects were then averaged to give an averaged fMRI intensity map. It should be noted that this fMRI intensity map reflects the temporal means of fMRI signals that were already removed by the standardization procedure when we derived the global co-activation patterns. Comparing the global co-activation map with this fMRI intensity map did not reveal any obvious spatial correspondence between de-activations and areas of low fMRI signal intensity (Supplementary Figure 3), suggesting the observed de-activations are not an artifact of a low fMRI signal intensity.

To estimate the proportion of the resting-state fMRI signal fluctuation that can be explained by the global component, the calculated global signal time course was correlated with every gray matter voxel and the resulting correlation was then squared to represent the percentage of temporal variation at each voxel explained by the global signal. The percentage values were then averaged across all gray matter voxels for each subject, and the mean and standard deviation across subjects were calculated and reported.

The global signal co-activation map of monkey fMRI data was created in a similar way, but the threshold for selecting time points with large global signal was relaxed to include the top 30% time points with the largest global signal (Supplementary Figure 2).

**Data availability**. The fMRI-LFP data from monkeys are available from the corresponding author upon reasonable request. The human fMRI data are available for download at the HCP website (https://www.humanconnectome.org).

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

## Acknowledgements

This research was supported (in part) by the NIH Intramural Research (ZIA MH002896, ZIC MH002899), and the NIH Pathway to Independence Award (K99/R00) 4R00NS092996-02. We thank Dr. Chiang-Shan Li at Yale University for sharing the brain mask for Ch4 neurons at the human basal forebrain.

## Author contributions

X.L. and J.H.D. designed the study; M.L.S., F.Q.Y. and D.A.L. conducted monkey experiments; X.L., J.A.d.Z., C.C. performed the analyses; X.L., J.A.d.Z., D.A.L., C.C. and J.H.D wrote the paper.

## Additional information

**Competing interests:** The authors declare no competing financial interests.

