## [Peer Review File · Nature Communications]

Reviewers' comments:

Reviewer #1 (Remarks to the Author):

This is an interesting observational study based on reanalysis of existing data, showing that resting fMRI signal increases in visual, sensory and motor cortex and to some extent other regions are positively correlated with a characteristic sequence of LFP changes (SSTs) in the monkey (as a follow-up to the previous electrophysiology-only study in the monkey from 2015 by the same group), and that resting fMRI signal increases in these same regions are correlated with fMRI decreases in several subcortical arousal areas in humans. The results are interesting and valuable to the field and methods are convincing. Specific comments, suggestions and limitations are below.

This study is purely correlative and therefore can only show associations not directions of causality. It is possible that the observed cortical changes are caused by subcortical ones, or vice-versa, or that both are caused by some other third driver. Statements such as "momentary drops in cortical arousal that appear orchestrated by the forebrain" and "widespread variations in fMRI cortical activity in part result from momentary arousal drops" in the Abstract and elsewhere imply causality, which has not been proven by the current study. Such statements should be modified or removed.

Results lines 165-166: "On average, the global signal variation explained a substantial proportion ($15.32\% \pm 3.69\%$) of the total signal variance seen in individual gray matter voxels." Details of how this was calculated should be provided.

Text reference to Fig2b mentions monkey data so should reference Fig2c instead.

Figure 2a legend should define what the threshold for "high-signal" was.

Preprocessing lines 334-337: 1 second window was used, but after removing 1.75 sec from the 2.6 data segment (TR = 2.6 sec) only 0.85 sec would remain. How can a 1 second window be used to analyze 0.85 second data segments?

An important potential limitation that should be discussed is the restriction of the HCP data to the 20% with the largest global fluctuations. It is possible that these are not representative of more typical brain function. I see now that there is a supplementary figure (S3) that aims to address this. But should be pointed out more clearly in the text.

Figure S4: The gray timecourse indicates "random control from basal forebrain" but where precisely does this timecourse come from (not indicated on figure)?

Lines 170-171 and 632-633. Need to explain why Z score >2.58 is statistically significant

Figure S2: large white arrows should be explained in the legend. Same holds for Figure S3 and S5.

Supplementary figures are not very well integrated into the main text and should each be briefly described where appropriate

Figure S5 middle rows figure legend should include additional information to explain what the gray scale percentages represent.

Figure S5 shows NB and thalamus don't colocalize with areas of reduced signal, but don't show the SN regions where fMRI decreases probably do colocalize with regions of reduced mean signal—this should be mentioned and/or added.

The Discussion section ends abruptly without even a brief concluding statement or statements; would suggest adding this.

Reviewer #2 (Remarks to the Author):

How to legitimately remove confounds from a time series of so-called 'resting-state' BOLD contrast fMRI images has long been a topic of controversy. Global normalization has been shown to be a source of spurious findings, such as non-existent anti-correlations between groups of brain areas. The problem has sharpened as the stability and sensitivity of MRI scanners have improved, so that signal fluctuations can now be mostly attributed to neurophysiological changes, head motion, cardiac cycle effects and respiration. It has also been long known, following the pioneering work of Shulman (van Eijsden 2009, not cited) that BOLD fMRI signal increases with increasing arousal. An extreme instance is found in the study of Salek-Haddadi, 2003, who found a very large negative BOLD signal everywhere in the cortex of a patient experiencing an absence seizure.

Thus it is plausible that global BOLD signal fluctuations in normal healthy primate brain may arise from temporal variations in arousal. The findings reported here are consistent with this partial explanation. The novel and striking observation that activity in the nucleus basalis is anti-correlated with widespread positive cortical BOLD fluctuations is highly significant. Such a 'see-saw' relationship with subcortical areas was found also by Portas in her 1998 fMRI study of the interaction of attention and arousal, and in the Salek-Haddadi study just mentioned, though in different subcortical areas.

The manuscript would be improved if discussion of these earlier findings were included, which might strengthen the rather speculative argument that the increased cortical BOLD signal results from a reduced inhibitory signal arising from a less-aroused nucleus basalis. Another improvement would be a succinct summary of the implications of the findings here reported for best practice in the analysis of resting state fMRI data. Should NB activity be used as a covariate in the analysis of the activity in the rest of the brain, to eliminate simple arousal effects? Or is it too soon to make this recommendation?

I recommend publication of this manuscript once these relatively minor issues have been dealt with.

Reviewer #1 (Remarks to the Author):

This is an interesting observational study based on reanalysis of existing data, showing that resting fMRI signal increases in visual, sensory and motor cortex and to some extent other regions are positively correlated with a characteristic sequence of LFP changes (SSTs) in the monkey (as a follow-up to the previous electrophysiology-only study in the monkey from 2015 by the same group), and that resting fMRI signal increases in these same regions are correlated with fMRI decreases in several subcortical arousal areas in humans. The results are interesting and valuable to the field and methods are convincing. Specific comments, suggestions and limitations are below.

Response:

Thank you for your constructive comments, which have helped us to improve the general quality of this paper significantly.

RIC1 (Reviewer 1 & Comment 1):

This study is purely correlative and therefore can only show associations not directions of causality. It is possible that the observed cortical changes are caused by subcortical ones, or vice-versa, or that both are caused by some other third driver. Statements such as “momentary drops in cortical arousal that appear orchestrated by the forebrain” and “widespread variations in fMRI cortical activity in part result from momentary arousal drops” in the Abstract and elsewhere imply causality, which has not been proven by the current study. Such statements should be modified or removed.

Response:

We completely agree with the reviewer about this point, particularly about the direction of causality between the cortical and subcortical changes. In fact, in our 2015 paper that first reported the SST (Liu, X., et al. Neuroimage, 2015) we have shown that the middle-frequency power reduction of the SST is more dominant at some high-order brain regions. These regions are largely overlapped with the areas that are known to project to the subcortical ascending systems (French, J.D. et al., Journal of Neurophysiology, 1955), suggesting a possibility that the cortical SST may initiate these cortical subcortical modulations.

Some of the statements in the original submission, including the two sentences quoted by the reviewer, are somewhat misleading and confusing. We’ve modified them in this revision. In addition, we added some discussion about the possibility that the cortical changes cause the subcortical modulations, and emphasized that our analysis is only correlative. Please see the third paragraph of the “Discussion” section for these changes.

Below are the specific changes we made in response to this point.

- 1. In the Abstract, we changed “momentary drops in cortical arousal that appear orchestrated by the basal forebrain” to “momentary drops in cortical arousal and are associated with activity changes in the basal forebrain and thalamus”.*
- 2. In the Abstract, we changed “widespread variations in fMRI cortical activity in part result from momentary arousal drops” to “widespread variations in fMRI cortical activity are associated with momentary arousal drops”.*

3. *In the first paragraph of the Discussion section, we changed “found further evidence for a contribution of arousal drops” to “found further evidence for a link to arousal drops”.*
4. *We added the following texts in the third paragraph of the Discussion section: “The potential role of the mid-frequency activity in mediating top-down inhibition also raises the possibility that the cortical changes in these arousal events precede the subcortical modulations, especially given that that mid-frequency power reduction has been shown to be stronger in brain regions that project to the subcortical nodes of the ascending arousal system. However, it should be emphasized that the presented analysis is purely based on correlations and does not permit one to infer a causal relationship between cortical and sub-cortical activity; this is a challenge that remains to be addressed in future studies.”*

R1C2:

Results lines 165-166: “On average, the global signal variation explained a substantial proportion (15.32%±3.69%) of the total signal variance seen in individual gray matter voxels.” Details of how this was calculated should be provided.

Response:

We have added into the “Methods” section (the second last paragraph) the detailed descriptions about how we obtained this percentage.

The new paragraph we added is that

“To estimate the proportion of the resting state fMRI signal fluctuation that can be explained by the global component, the calculated global signal time course was correlated with every gray matter voxel and the resulting correlation was then squared to represent the percentage of temporal variation at each voxel explained by the global signal. The percentage values were then averaged across all gray matter voxels for each subject, and the mean and standard deviation across subjects were calculated and reported.”

R1C3:

Text reference to Fig2b mentions monkey data so should reference Fig2c instead.

Response:

We have corrected this and other related errors in the revision, and thank the reviewer for pointing it out.

Specifically, in the fourth paragraph of the Results section, we changed “Fig. 2a” to “Fig. 2b”, and “Fig. 2b” to “Fig. 2a”. In addition, we also added “(Fig. 2a)” at the end of the first sentence.

R1C4:

Figure 2a legend should define what the threshold for “high-signal” was.

Response:

We have described how we defined the high-signal in the figure 2a caption in the revision.

Specifically, in the caption of Figure 2 we changed “the high-signal (red) time points.” to “the high-signal (red) time points, which were defined to have a global signal larger than 0.2 and made up 16.6% of total number of time points.”

R1C5:

Preprocessing lines 334-337: 1 second window was used, but after removing 1.75 sec from the 2.6 data segment (TR = 2.6 sec) only 0.85 sec would remain. How can a 1 second window be used to analyze 0.85 second data segments?

Response:

The reviewer is right that only 0.85 s of LFP data was analyzed per MRI TR (2.6 s). The 1 second window refers to the multi-taper method, which was applied to the concatenated time series of 0.85s segments of LFP data. After time-frequency analysis, the spectrograms were temporally rescaled to the 2.6 s resolution, effectively blurring over the gaps of missing LFP data. This type of analysis allowed for interpolation of temporal dimension of the spectrograms; of course, it did not improve its inherent temporal resolution, which remained that of the MRI TR (2.6 seconds). To make sure this approach was appropriate and did not cause unintentional distortion of the spectrograms, we compared it with using the full 2.6 seconds of LFP data acquired during each MRI TR. For this purpose we removed MRI-gradient-induced artifacts using a custom algorithm (Liu, Z. et al, NeuroImage 2012) that we routinely use for human fMRI-EEG data processing. The results derived by this approach using all the data are very similar to those derived based on these 0.85-sec concatenated segments (See the figure below for an example from monkey A). However, the algorithm failed to completely remove the artifacts and the remaining residuals caused some oscillating patterns at the higher frequencies (>40 Hz) of the spectrogram, so we decided to only show the results from the concatenation approach. We want to emphasize that these two approaches gave very similar SST patterns and identical general findings and conclusions of our study.

The gap method using only
0.85-sec data of each TR

The artifact-removal
method using all the data

R1C6:

An important potential limitation that should be discussed is the restriction of the HCP data to the 20% with the largest global fluctuations. It is possible that these are not representative of more typical brain function. I see now that there is a supplementary figure (S3) that aims to address this. But should be pointed out more clearly in the text.

Response:

In the revision, we added some discussion about this when introducing Figure S3 (Supplementary Figure 4 in the revision). Please refer to the last paragraph of the “Result” section for the modifications we made regarding this comment.

Specifically, we changed

“While these findings reported above were obtained from a subset of the HCP database, they were highly reproducible across sessions and days, across randomly selected subgroups of subjects, and with much smaller numbers of subjects (Fig. S3)”

to

“It should be noted that the findings reported above were obtained from only 20% of the subjects showing the largest global signal fluctuations, and thus may not be typical of the general population. This concern is alleviated, however, by the fact that the findings were reproduced by analyzing the data across sessions and days, from randomly selected subgroups of subjects, and from much smaller numbers of subjects (Supplementary Fig. 4).”

R1C7:

Figure S4: The gray timecourse indicates “random control from basal forebrain” but where precisely does this timecourse come from (not indicated on figure)?

Response:

The “random control from basal forebrain” is the average of randomly selected segments of fMRI signals from the basal forebrain mask. We’ve added this information in the caption of Figure S4 (Supplementary Figure 3 in the revision) in the revision.

Specifically, the legend for the gray time course has been changed from “random control from basal forebrain” to “signal from the basal forebrain not aligned to global peaks”

R1C8:

Lines 170-171 and 632-633. Need to explain why Z score >2.58 is statistically significant

Response:

The Z score of 2.58 is equivalent to a p-value of 0.01. In the revision, we have added this information at the two places pointed out by the reviewer.

Below are the specific changes we made in response to this point.

- 1. In the fourth paragraph of the Results section, we changed “(these had Z scores > 2.58)” to “(these had Z scores > 2.58, which corresponded to a significance p-value < 0.01)”.*
- 2. In the caption of Figure 2, we changed “significant score (Z > 2.58)” to “significant score (Z > 2.58, equivalent to p < 0.01)”.*

R1C9:

Figure S2: large white arrows should be explained in the legend. Same holds for Figure S3 and S5.

Response:

We have added the corresponding descriptions of these white arrows in the captions of Figure S2, S3, and S5 (Supplementary Figure 2, 4, and 5 in the revision).

Below are the specific changes we made in response to this point.

- 1. In the caption of Supplementary Figure 2, we added “The white arrows indicate the de-activated regions around the brain areas homologous to the basal forebrain of the human brain.”*
- 2. In the caption of Supplementary Figure 4, we added “The white arrows in the top, middle, and bottom rows indicate de-activations in the nucleus basalis (NB) of the basal forebrain, the dorsal midline thalamus, and the midbrain structure respectively. ”*
- 3. In the caption of Supplementary Figure 5, we added “The white arrows in (a), (b), and (c) point to the de-activated regions in the nucleus basalis (NB) of the basal forebrain, the dorsal midline thalamus, and the midbrain structure, respectively.”*

R1C10:

Supplementary figures are not very well integrated into the main text and should each be briefly described where appropriate

Response:

We have reordered the supplementary figures, and briefly introduced them at the main text where it is appropriate.

Below are the specific changes we made in response to this point.

- 1. At the end of the fifth paragraph of the Results section, we added “Of note, fMRI co-activation patterns derived from the monkey fMRI data showed a similar spatial distribution (Supplementary Fig. 2), although the poorer spatial resolution prevented a detailed comparison.”*
- 2. We added the sixth paragraph of the Results section to introduce the Supplementary Figure 3, which is as “Lastly, we investigated whether the temporal scale of co-activations in the HCP data was consistent with that of the SST. For that purpose, we averaged the fMRI signal around the occurrence of global peaks at various time lags. It was observed that co-activations spanned about 10 seconds, consistent with the approximate duration of an SST (Supplementary Fig. 3).”*
- 3. We changed the last paragraph of the Results section to introduce the Supplementary Figure 4 and 5, which is from
“While these findings reported above were obtained from a subset of the HCP database, they were highly reproducible across sessions and days, across randomly selected*

*subgroups of subjects, and with much smaller numbers of subjects (Fig. S3).”
to*

“It should be noted that the findings reported above were obtained from only 20% of the subjects showing the largest global signal fluctuations, and thus may not be typical of the general population. This concern is alleviated, however, by the fact that the findings were reproduced by analyzing the data across sessions and days, from randomly selected subgroups of subjects, and from much smaller numbers of subjects (Supplementary Fig. 4). Another potential concern is that the apparent subcortical de-activations were an accidental result due to weak fMRI signals in these regions associated with magnetic field inhomogeneity; however, this is unlikely since the absolute fMRI signal has been removed from the analysis during pre-processing and the co-(de)activations quantify relative changes only. In addition, the pattern of global co-(de)activations and mean fMRI signal strength (Supplementary Fig. 5) appear to have little in common.”.

R1C11:

Figure S5 middle rows figure legend should include additional information to explain what the gray scale percentages represent.

Response:

The gray scale percentages are relatively to the average across all the brain voxels. We have added that information in the caption of Figure S5 (Supplementary Figure 5 in the revision).

We added the following sentence into the caption of Supplementary Figure 5: “The fMRI intensity map (middle row) is shown as the percentage of the overall mean across the whole brain.”

R1C12:

Figure S5 shows NB and thalamus don't colocalize with areas of reduced signal, but don't show the SN regions where fMRI decreases probably do colocalize with regions of reduced mean signal—this should be mentioned and/or added.

Response:

We agree with the reviewer that the midbrain structure, which we tentatively regarded as the SN region, showed weaker fMRI signal in the mean intensity map. But we also noticed that the adjacent voxels, which also have lower fMRI intensities, do not show significant de-activations at these global fMRI peaks. We've added this information to the caption of Figure S5 (Supplementary Figure 5 in the revision).

We have added the following sentence into the caption of Supplementary Figure 5:

“The de-activated voxels in the midbrain structure did show lower mean fMRI signal, but adjacent voxels that also have weak mean fMRI signal show no de-activation at all, especially in a structure right above the de-activated region.”

R1C13:

The Discussion section ends abruptly without even a brief concluding statement or statements; would suggest adding this.

Response:

This is a very good suggestion. We have now added a summary paragraph at the end of the Discussion section, which goes as

“To summarize, this study demonstrated a tight coupling between a temporally-distinct electrophysiological event, the SST, and widespread fMRI activity whose spatial and temporal characteristics are consistent with an origin in momentary drops in arousal. The presented findings provide novel insight into the origin of the global fMRI signal and its relationship with arousal, paving the way for methods that properly account for arousal fluctuations during resting state fMRI.”

Reviewer #2 (Remarks to the Author):

R2C1:

How to legitimately remove confounds from a time series of so-called 'resting-state' BOLD contrast fMRI images has long been a topic of controversy. Global normalization has been shown to be a source of spurious findings, such as non-existent anti-correlations between groups of brain areas. The problem has sharpened as the stability and sensitivity of MRI scanners have improved, so that signal fluctuations can now be mostly attributed to neurophysiological changes, head motion, cardiac cycle effects and respiration. It has also been long known, following the pioneering work of Shulman (van Eijsden 2009, not cited) that BOLD fMRI signal increases with increasing arousal. An extreme instance is found in the study of Salek-Haddadi, 2003, who found a very large negative BOLD signal everywhere in the cortex of a patient experiencing an absence seizure.

Thus it is plausible that global BOLD signal fluctuations in normal healthy primate brain may arise from temporal variations in arousal. The findings reported here are consistent with this partial explanation. The novel and striking observation that activity in the nucleus basalis is anti-correlated with widespread positive cortical BOLD fluctuations is highly significant. Such a 'see-saw' relationship with subcortical areas was found also by Portas in her 1998 fMRI study of the interaction of attention and arousal, and in the Salek-Haddadi study just mentioned, though in different subcortical areas.

The manuscript would be improved if discussion of these earlier findings were included, which might strengthen the rather speculative argument that the increased cortical BOLD signal results from a reduced inhibitory signal arising from a less-aroused nucleus basalis.

Response:

We thank the reviewer for the constructive comments and pointing us to these publications, which are closely related to the observation in our study. We added two new paragraphs to discuss about them in the “Discussion” section (the fourth and fifth paragraphs).

In particular, the “global” fMRI responses in rats (van Eijsden, P., et al., Neuroimage, 2009; Maandag, N.J.G. et al. PNAS, 2007) may be related to the anesthetized condition showing a burst-suppression EEG pattern; this is because the 1% halothane used in that study is expected to induce such a EEG pattern in rats. In fact, in previous investigations, we have observed the global rsfMRI signal and spatially non-specific rsfMRI correlations under the burst-suppression anesthesia, and suspected that the generation of such EEG bursts may involve similar mechanisms or pathways as the SSTs observed here. For this reason, we also included references to some of our own previous studies in the new paragraph.

Below are the two paragraphs we added into the Discussion section to address this comment: “Widespread fMRI signal fluctuations have been observed previously and related to changes in brain arousal state, and both cortical and subcortical involvement have been reported. For example, a previous neuroimaging study found a specific role of the thalamus in mediating the interaction of selective attention and arousal⁴⁵. Furthermore, concurrent fMRI-EEG recordings in human epilepsy have shown that absence seizures accompanied by generalized spike-wave discharge (GSWD) activity, are associated with wide-spread fMRI signal decreases in the cortex and focal increases in the median portion of the thalamus⁴⁰. While this anti-polar cortical-subcortical relationship was also seen in our study, it is not clear how the difference in the direction of changes between the studies can be reconciled. It is possible that the sustained (up to 1 minute) low-frequency (~3 Hz) GSWD is similar in nature to neural activity during the late stage of the SST, where fMRI signal polarity appears to reverse (Supplementary Fig. 3) and becomes consistent with that seen in epilepsy.

In animals, anesthetic conditions characterized by a burst-suppression EEG pattern have also shown wide-spread fMRI signal fluctuations. These co-occur with EEG bursts that either present spontaneously⁴¹ or are elicited by sensory stimulations^{42–44}, and appear somewhat stronger at sensory regions^{43,44}. It is, however, unclear whether the EEG bursts during anesthesia and the SSTs in the drowsy state are generated by similar mechanisms. In this regard, it should be realized that the coupling between the local electrophysiological signal, neuronal energy consumption, and the fMRI response is complex and may vary across arousal states⁴⁴. ”

R2C2:

Another improvement would be a succinct summary of the implications of the findings here reported for best practice in the analysis of resting state fMRI data. Should NB activity be used as a covariate in the analysis of the activity in the rest of the brain, to eliminate simple arousal effects? Or is it too soon to make this recommendation?

Response:

We agree and have added some discussion as the second last paragraph of the “Discussion” section. Other than avoiding transitions in arousal state by better control of the experimental conditions, removal of arousal effects may benefit from taking into account the spatial patterns of the global co-activations, e.g., the sensory dominance and subcortical deactivations. For this purpose, we would recommend to use a spatial co-activation template, either from this study or the one derived previously from fMRI correlations to an EEG-based arousal index (Chang, C. et al. PNAS, 2016), to identify and then remove these affected time points. Alternatively, it may be possible to identify arousal transitions from other physiological signals acquired from the periphery (e.g. cardio-vascular variations). In fact, we have preliminary data (but no solid recommendation yet) that this can be quite effective.

The new paragraph we added into the Discussion section to address this point is as below: “Given a strong influence of arousal state on the fMRI signal, and the difficulty in minimizing arousal variations, the most fruitful way to reduce confounding effects on the interpretation of resting state data may be to remove or account for them in post-processing. A simple strategy is to remove the global signal component given its close relationship to arousal fluctuations, and

multiple approaches have been proposed to achieve this^{55–57}. While this strategy has been controversial^{58,59}, some consensus appears to be developing on whether and how it should be done⁴⁷. Caution should be exercised when applying this method to resting state fMRI datasets without much global signal changes, since it can potentially remove other types of fluctuations not related to arousal. An alternative strategy would be to identify and then exclude time points showing spatio-temporal patterns similar to either that of global co-activations we observed in this study or to the one derived based on fMRI correlations with an EEG-based arousal index⁹. This second strategy may be most specific to arousal effects and have less unintended effects on the data.”

REVIEWERS' COMMENTS:

Reviewer #1 (Remarks to the Author):

The authors have addressed all comments.

Minor remaining revision needed for preprocessing lines 403-404: The authors have clarified my question about the 1 second window for 0.85 s data segments in their response to reviewers, but to help other readers it would be helpful if the authors could also briefly clarify in the methods that the 1 second window was applied to concatenated data segments for the multi-taper analysis. For example, they could modify lines 403-404 to say something like "was discarded to remove signals contaminated with MR-gradient induced artifacts and the remaining 0.85 sec data segments were concatenated for further analysis."

Reviewer #2 (Remarks to the Author):

The authors have responded appropriately to my comments. In my view, the manuscript is ready for publication.

Reviewer #1 (Remarks to the Author):

The authors have addressed all comments.

Minor remaining revision needed for preprocessing lines 403-404: The authors have clarified my question about the 1 second window for 0.85 s data segments in their response to reviewers, but to help other readers it would be helpful if the authors could also briefly clarify in the methods that the 1 second window was applied to concatenated data segments for the multi-taper analysis. For example, they could modify lines 403-404 to say something like “was discarded to remove signals contaminated with MR-gradient induced artifacts and the remaining 0.85 sec data segments were concatenated for further analysis.”

Response:

We modified the text exactly following the suggestion by the reviewer. Please refer to the first paragraph in the Preprocessing subsection of the Methods section for the changes.

Reviewer #2 (Remarks to the Author):

The authors have responded appropriately to my comments. In my view, the manuscript is ready for publication.